# Rosemary Leaf Extract Inhibits Glycation, Breast Cancer Proliferation, and Diabetes Risks

**Yixiao Shen [1,2,†], Jing Han [1,3,†], Lili Zheng [1], Xiaoyan Zheng [1], Binling Ai [1], Yang Yang [1], Dao Xiao [1] and Zhanwu Sheng [1,*]**

1.  Haikou Experimental Station, Chinese Academy of Tropical Agricultural Sciences, Haikou 570101, China; Lily_0909@126.com (L.Z.); zhxyhn@126.com (X.Z.); aibinling@163.com (B.A.); yangyang19@catas.cn (Y.Y.); xd_dao35@126.com (D.X.)
2.  College of Food Science, Shenyang Agricultural University, Shenyang 110161, China; shenyixiao87@126.com
3.  Gansu Cheezheng Industrial Group Co., Ltd, Gansu 730010, China; hjz0822@qzh.cn
*   Correspondence: zsheng@catas.cn
†   The authors have contributed equally to this manuscript.

**Abstract:** Advanced glycation end products (AGEs) generated from glycation can cause inflammation-related diseases such as diabetes and cancer. The bioactive compounds of rosemary extract (RE) were extracted and incubated with sugar-protein rich food and breast cancer cell MCF-7 to investigate its inhibitory effect on glycation and cancer cell proliferation, respectively. The diabetic rat was dosed with RE to investigate its effect on blood glucose, serum malondialdehyde (MDA), cholesterol (CHO), triglycerides (TG), low-density lipoproteins (LDLs), anti-oxidation capacity (T-AOC), superoxide dismutase (SOD) activity, anti-oxidation capacity alkaline phosphatase (ALP), glutamate pyruvate transaminase (GPT), and glutamate oxaloacetate transaminase (GOT). The results show that RE contained seven major phenolics ranging from 17.82 mg/g for rosemarinic acid to 0.01 mg/g for ferulic acid on dry weight basis. It significantly lowered AGEs, carboxymethyl lysine (CML), and protein glycation in a sugar-protein rich intermediate-moisture-food (IMF) model. Furthermore, the survival rates of MCF-7 cells decreased to 6.02 and 2.16 % after 96 h of incubation with 1.0 and 2.0 mg/mL of RE, respectively. The blood glucose, MDA, CHO, TG, and LDLs in diabetic rats of RE treatment were decreased. The RE treatment also enhanced the T-AOC and SOD activity. Furthermore, the RE treatment improved liver function through improving ALP, GPT, and GOT activities in diabetic rats. The results provide important information for the nutriaceutical and pharmaceutical application of rosemary extract.

**Keywords:** rosemary; glycation; cancer; diabetes; phenolics

## 1. Introduction

Glycation is a non-enzymatic reaction that takes place between reducing sugars and the free amino residues of proteins [1]. By altering the protein conformation and inducing protein aggregation and cross-linking, glycation eventually forms a group of heterogeneous macromolecules, characterized as "advanced glycation end products" (AGEs) [2]. Accumulating evidences indicate that AGEs could promote the initiation and progression of carcinogenesis via inducing oxidative stress and accelerating the progression of inflammation-related diseases [3]. The pathological roles of AGEs in diabetes and cancer has also been garnered [4–5]. For example, diabetes mellitus (DM) is a group of metabolic disorders resulting from pancreatic insulin secretion deficiency or ineffectiveness [6]. It can cause hyperglycaemia and abnormalities in the carbohydrate, fat, and protein metabolism [7]. Thus, the levels of blood glucose, oxidative stress, and liver function are considered as the

primary symbol of DM and believed to be the important factors in the development of diabetic complications. Many epidemiological researches have concluded that some cancers are usually associated with diabetes [8–9]. Patients with DM have an increased likelihood of developing various types of cancers, especially breast cancer through AGE formation, which may lead to cellular and bio-molecular dysfunction [10].

Rosemary (Rosmarinus officinalis L., Lamiaceae) is a perennial shrubby herb that has its origins in the Mediterranean region [11]. Nowadays, it has been cultivated worldwide as a common ornamental and aromatic herb plant [12]. Furthermore, rosemary leaf extract is widely used as a preservative in the food industry, due to its abundant antioxidant compounds [13]. Rosemary leaf extract is also highlighted to have pharmacological activities such as anti-inflammatory, anti-obesity, and antimicrobial activity, which are attributed to its bioactive composition [12]. AGEs are not only inducers of diabetes but also associated with the development of cancer. Although the anti-proliferative activity of rosemary against different breast cancer cell lines has been studied, a comprehensive study of rosemary extract antiglycation capability and anticancer potential has not been well documented [14]. Therefore, in this study, the main phenolic compounds in rosemary leaf extract (RE) are identified and quantified. Then, the abilities of RE to inhibiting protein glycation, AGEs and carboxymethyl lysine (CML) formation in sugar-protein rich intermediate-moisture food (IMFs) are evaluated. Furthermore, the inhibitory effects of RE on non-invasive hormone-dependent breast cancer cell (MCF-7) proliferation and risk of diabetic rats are investigated. The results provide important information for nutriaceutical and pharmaceutical application of rosemary extract.

## 2. Materials and Methods

### 2.1. Chemicals and materials

Protocatechuic acid, caffeic acid, ellagic acid, ferulic acid, rosemarinic acid, carnosol and carnosic acid standards, sodium chloride, sodium bromide, glucose, glycerol, streptozotocin (STZ), sodium azide, trichloroacetic acid, phosphate buffer, thionyl chloride, dichloromethane, and trifluoroacetic acid anhydride were purchased from Sigma-Aldrich (St. Louis, MO, USA). HPLC grade methanol, ethanol, acetonitrile, chloroform, and acetone, as well as ACS grade acetic acid and hydrochloric acid were obtained from Thermo Fisher Scientific Co. (Pittsburg, PA, USA). Whey protein isolate (WPI) and alcalase were purchased from Davisco Foods International, Inc. (Eden Prairie, MN) and Novozymes A/S (Bagsvaerd, Denmark), respectively. Nε-(Carboxymethyl)-L-lysine (CML) standard was supplied by Toronto Research Chemicals Inc. (Toronto, Ontario, Canada).

### 2.2. Preparation of rosemary extract

Fresh rosemary leaves were ground after they were harvested from a local farm (Haikou, China). Then, 100 g of the ground rosemary leaves were extracted by 400 mL of ethanol for 24 h on the basis of the method of Bokaeian with minor modification [15]. The solvent was separated by filtration and evaporated by a vacuum centrifuge evaporator (Labconco, Kansas City, MO, USA) [15]. The dried rosemary extract was stored at –20°C until use. A stock extract solution at a concentration of 2.0 mg/mL were prepared in deionized water for the determination of the phenolics.

### 2.3. Determination of phenolics in rosemary extract

One milliliter of rosemary extract solution (2 mg/mL) was transferred to an HPLC vial and determined using a Waters 2690 high-performance liquid chromatography system (Milford, MA, USA) with a photodiode array detector and a C18 column (id 250 × 4.60 mm 5 micron) (Phenomenex, Torrance, CA, USA). The HPLC analysis condition was based on the study of Shen et al [16]. The phenolics were quantified by using the calibration curves of their corresponding standards.

*2.4. Determination of AGEs, CML, and protein glycation in protein-sugar rich intermediate-moisture food*

2.4.1. Preparation of protein−sugar rich intermediate-moisture foods

The intermediate-moisture food (IMF) was prepared according to the study of Sheng et al [17]. The control IMF dough (200 g) consisted of 45% of WPI, 30% of glycerol, 12.5% of glucose, and 12.5% of deionized water. They were mixed homogenously to form a dough after sodium azide (0.08 g) was added in for preventing microbial growth. Three rosemary extract fortified IMF doughs contained 1%, 2%, and 4% of rosemary extract, respectively. Control or fortified IMF dough was placed on a Petri dish laid on a rack in an airtight plastic box under two different water activity (wa) conditions. The wa 0.75 was prepared by saturated sodium chloride solution and wa 0.56 was prepared by sodium bromide solution. All of the boxes were capped and incubated at 45°C. Ten grams of each dough were sampled at different storage times for determining AGEs and protein glycation.

2.4.2. Measurement of AGEs

The control or fortified IMF sample (500 mg) was dissolved in double distilled water (10 mL) with magnetic stirring at room temperature for 80 min. Then, the solution was centrifuged at 4000 xg for 30 min and an aliquot of 4 mL supernatant was collected. The fluorescence intensity of the supernatant was recorded by an F-4500 Luminescence Spectrometer (Shimadzu, Japan). The excitation wavelength was set at 370 nm and an emission wavelength was set at 440 nm with a slit width of 5 nm. The intensity was used to express as the level of AGEs in the sample.

2.4.3. Measurement of CML

The level of CML was determined by the GC-MS method based on a previous study of Sheng et al with minor modification [17]. Two hundred milligrams of control or fortified IMF sample was mixed with 20 mL of chloroform/acetone solvent (1:3, v:v) for 10 min. Then, the mixture was centrifuged at 4000 xg for 15 min for separating the fat-solubilized organic upper layer. The precipitated protein was dried and hydrolyzed in 8 mL of hydrochloric acid solution (6 N) at 110°C for 24 h. An aliquot of 50 μL protein hydrolysate was dissolved in double distilled water (1.0 mL). After being filtered and dried, the protein hydrolysate was reacted with 1 mL of thionyl chloride/methanol (v: v, 1.46:100) at 110°C for 30 min and then dried again. The derivatization of the dried sample was carried out by mixing with 2 mL of dichloromethane and 400 μL of trifluoroacetic acid anhydride at room temperature for 1 h. The sample was analyzed by GC-MS (Agilent 7890B, 7693 mass selective detector single quadrupole mass spectrometer system) coupled with an HP-5MS column (30m × 0.25 mm × 0.25 μm, Palo Alto, CA). The GC condition is described as follows: oven temperature was held at 40°C for 1 min and increased to 70°C at 20°C/min followed by ramping to 300°C at 50°C/min and holding at 300°C for 2 min. The carrier gas was helium and the flow rate was set at 1.20 mL/min. The MS ion source temperature and transfer line temperature were set at 230°C and 250°C, respectively. The electron ionization (EI) mode was used for MS operation with electron energy at 70 eV and an ion scan range of m/z 40−800. A calibration curve of CML was used for quantification.

2.4.4. Measurement of protein glycation

The degree of glycated protein was determined according to the method described in a previous study [17]. Three hundred micrograms of control or fortified IMF was dissolved in 15 mL of double-distilled water and stirred at room temperature for 80 min. After being centrifuged at 4000 xg for 30 min, 100 μL of the supernatant was taken and diluted 10 times with double distilled water and then filtered. The filtered sample was analyzed by a LC-MS system with a Waters UPLC ZMD 4000 (Waters Co., Milford, MA) and TOF mass spectrometer. The LC-MS analysis conditions were the same as in the study of Sheng et al [17].

*2.5. Cell culture of MCF-7 cells and determination of cell proliferation and apoptosis*

The human breast cancer cell line (MCF-7) was procured from the Cell Bank of Shanghai Institutes for Biological Sciences (Shanghai, China). The cells cultured in Dulbecco's modified Eagle's medium (DMEM) media contained 10% fetal bovine serum (FBS), 100 U/mL penicillin, and 100 U/mL streptomycin. The culture plates were incubated at 37°C under an atmosphere of 5% carbon dioxide and 95% air. Then MCF-7 cells were inoculated at a density of $3 \times 10^3$ cells/well into 96-well-plates and incubated for 18 h at 37°C. After the medium was removed, the cells were washed by phosphate buffer solution (PBS). An aliquot of 100 μL of fresh medium containing rosemary extract at a concentration of 0, 0.125, 0.250, 0.500, 1.000, or 2.000 mg/mL was added and then incubated for 24 h at 37°C. The fresh medium without the addition of rosemary extract was used as the control. The number of cells that survived in each well was determined by using cck-8 kit (Yeasen Company, Shanghai, China). The absorbance of each well was then measured at 450 nm. The survival rate was calculated by the formula given below.

$$Survival = \frac{(Atrt - Ablk)}{(Actrl - Ablk)}$$

Atrt = absorbance of treatment cell well; Ablk = absorbance of medium and cck-8 without cell; Actrl = absorbance of control cell well.

Quantification assay of apoptotic cells was carried out by using the TdT-medicated dUTP Nick-End Labeling (TUNEL) Kit (Yeasen Company, Shanghai, China). After being incubated at DMEM medium, MCF-7 cells were washed twice with PBS solution and diluted with PBS to a density of $2 \times 10^7$ cells/mL. An aliquot of 500 μL of the diluted MCF-7 cells suspension was loaded on a poly-L-lysine coated slide and incubated in 4% paraformaldehyde in PBS for 15 min at 4°C. The sample slide was then washed twice with PBS. Then, the cell-loaded area was treated by 34 μL of double distilled water, 10 μL of equilibration buffer, 5 μL of FITC-12-dUTP labeling mixture, and 1 μL of recombinant TdT enzyme and then covered with a coverslip. After incubation at 37°C for 60 min in the dark, the coverslip was removed and the slide was incubated in PBS for 5 min. The slide was then incubated in 4′, 6-diamidino-2-phenylindole (DAPI) solution for 5 min at room temperature in the dark. Finally, the slides were immersed in deionized water to rinse. The apoptotic cells on the slide exhibited a strong green fluorescence which was monitored under a standard fluorescein filter at 520 nm. The total amount of cells (DAPI-stained nuclei) with blue color was detected at 460 nm. Eight areas on the slide were randomly selected and recorded under 200 times magnification level by using an upright fluorescence microscope (Zeiss, München, Germany). The ratio of apoptotic cells to total cells was calculated by the number of cells with green fluorescence divided by the number of cells with blue fluorescence to express cell apoptosis rate values in treatment or control group.

*2.6. Animal experiment and blood sample analysis*

Healthy male Sprague-Dawley (origin) rats weighing 190 ± 10 g were used in the animal experiment. Twenty-four rats were divided into three groups with eight in each group as normal control (NC), diabetic control (DC), and diabetic group with rosemary extract treatment (DRE) groups. All the rats were caged under the conditions of humidity (55 ± 5%), temperature (at 23°C), and light (12/12 h light/dark cycle) with free access to food and water. After being fed with a basic diet for one week, the rats were fasted for 12 h. The rats in the DC and RE groups were treated by intraperitoneal injection of streptozotocin (STZ) at a dose of 45 mg/kg body weight to induce hyperglycemia stress [18]. The rats in the NC and DC groups were still fed with the basal diet for 6 weeks. The rats in the RE group were administered for receiving rosemary extract based on 0.2 g/kg of body weight through oral gavage each day for the 6 consecutive weeks.

Blood was then collected from the rat's tail vein in order to measure the concentration of glucose weekly by using a glucose assay kit (GlucoDr, Germany). After 6 weeks, the animals were all sacrificed and blood samples were drawn by heart puncture. After being centrifuged at 1500 g for 10 min, the serum of the blood sample was separated and stored for further analysis. The concentrations of low density lipoprotein (LDL), high density lipoprotein (HDL), triglycerides (TG), cholesterol

(CHO), and the activities of liver enzymes including alkaline phosphatase (ALP), glutamic oxaloacetic transaminase (GOT) and glutamic pyruvic transaminase (GPT), as well as antioxidant enzyme superoxide dismutase (SOD), anti-oxidation capacity (AOC), malonaldehyde were determined by commercial assay kits. The procedures in the animal experiment were approved and complied with the Chinese Code of Practice for the Care and Use of Animals for Scientific Purposes.

### 2.7. Data analysis

Each determination was repeated with three replications. The values were expressed as the mean ± standard deviation. The data were analyzed by ANOVA with General Linear Model procedure (SAS system, SAS Institute Inc., Cary, NC) and the significant differences between means were at $P < 0.05$. The cell viability was repeated five times and analyzed by GraphPad Prism (version 6.0; GraphPad Software Inc., La Jolla, CA, USA).

## 3. Results and Discussion

### 3.1. Major phenolics in rosemary extract

In this study, seven major phenolics including protocatechuic acid, caffeic acid, ellagic acid, ferulic acid, rosemarinic acid, carnosol, and carnosic acid were identified in rosemary leaves extract (Figure 1 and Table 1). Rosemarinic acid was the dominant phenolic with a concentration of 17.28 ± 0.74 mg/g DW followed by carnosic acid (8.31 ± 0.06 mg/g DW) and carnosol (4.45 ± 0.03 mg/g DW). The concentrations of the other four phenolics were in a range of 2.16 ± 0.01 mg/g DW for protocatechuic acid to 0.01 ± 0.00 mg/g DW for ferulic acid (Table 1). Rosemary, together with mint, sage, marjoram, oregano, and thyme all belong to Lamiaceae family. Compared with oregano (6.61 ± 0.30 mg/g DW) and thyme (3.37 ± 0.01 mg/g DW) in the study of Hossain, as well as mint (1.45 to 4.16 mg/g) in the study of Castada, rosemary extract had the highest level of rosemarinic acid [19,20]. The carnosic acid in rosemary extract were significantly higher than oregano (4.72 ± 0.09 mg/g DW), marjoram (3.01 ± 0.05 mg/g DW), and thyme (6.41 ± 0.09 mg/g; DW) as well [19]. Furthermore, the carnosol was approximately three time higher than marjoram (1.76 ± 0.07 mg/g DW) and basil (1.38 ± 0.07 mg/g DW) [19]. It has been reported that RA is able to attenuate allergic diseases and slow the development of Alzheimer's disease [21]. Besides, carnosic acid and carnosol can protect thylakoid membranes against lipid peroxidation [22].

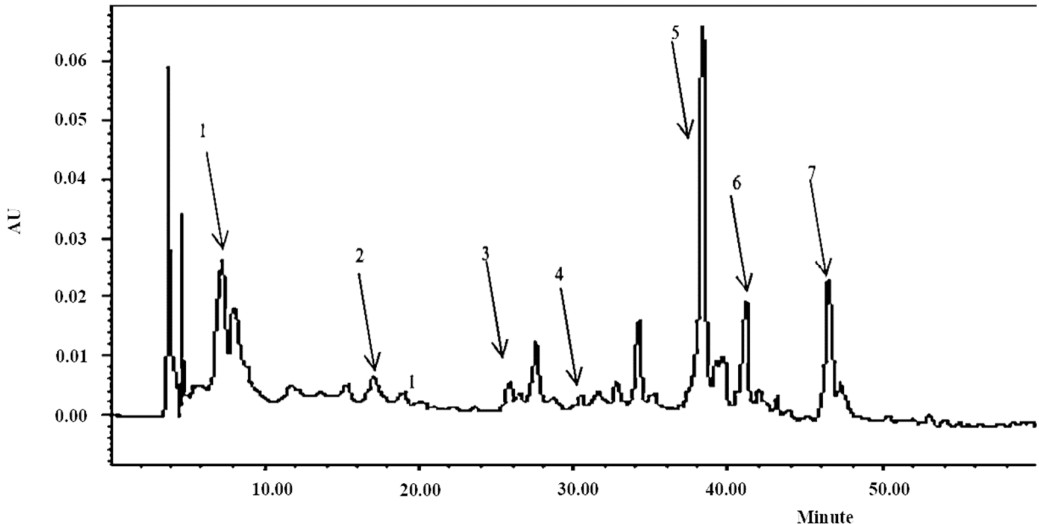

**Figure 1.** Chromatogram of rosemary extract under the wavelength of 280 nm:1. Protocatechuic acid (8.5 min); 2. caffeic acid (17.2 min); 3. ellagic acid (27 min); 4. ferulic acid (30 min); 5. rosemarinic acid (38 min); 6. carnosol (41 min); 7. carnosic acid (47 min).

**Table 1.** Major phenolic compounds in rosemary extract.

| Peak No. | Compounds | Concentration (mg/g DW) |
|---|---|---|
| 1 | Protocatechuic acid | $2.16 \pm 0.01$ |
| 2 | Caffeic acid | $0.02 \pm 0.00$ |
| 3 | Ellagic acid | $2.04 \pm 0.01$ |
| 4 | Ferulic acid | $0.01 \pm 0.00$ |
| 5 | Rosemarinic acid | $17.28 \pm 0.74$ |
| 6 | Carnosol | $4.45 \pm 0.03$ |
| 7 | Carnosic acid | $8.31 \pm 0.06$ |

DW: Dry weight basis

### 3.2. Effect of rosemary extract on AGEs, CML and glycation in sugar-protein rich foods

AGEs are a group of heterogeneous compounds generated through the Maillard reaction [23]. Generally, they can be cataloged into fluorescent cross-linking AGEs and non-fluorescent non-cross-linking AGEs (such as CML) on the basis of the fluorescence properties and cross-linking structures [23]. Covalently cross-linked protein aggregates are reported to be involved in the development of chronic diabetic complications [24]. The intake of AGEs induces intracellular oxidant stress by the interruption of the calorie restriction on oxidant stress and causes diabetes mellitus or other chronic diseases in biological systems [23]. Therefore, the inhibition of AGEs, CML, and protein glycation formation in both the biological and food systems is critical for reducing the risks of age-related diseases. In this study, the inhibitory efficiency of rosemary extract on the formation of AGEs and CML as well as protein glycation were evaluated by a sugar-protein rich food model.

As shown in Figure 2a, the levels of fluorescent intensity representing AGEs of control and all treatments decreased in the first 14 day but increased after day 21. It was because during the early storage of high protein food, amino acids and glucose would form glycosylation products, most of which had no fluorescence absorbance [24]. Furthermore, the reduce of some amino acids with fluorescence property by the Maillard reaction caused the decrement of fluorescence intensity [24]. However, after 21 days, the reaction could further react with free amino groups and thiol groups of protein, forming stable AGEs products. Thus, the fluorescence at the late stage dramatically increased [24]. Two water activities of wa 0.75 and wa 0.56 were used in this study to represent the most common values in food. A water activity higher than 0.80 would shift the reaction equilibrium from the glycated proteins end to the reactants end, which limits the reactants molecular mobility. As shown in Figure 2, the same trend of AGEs formation was observed in the IMF under both the wa 0.75 and 0.56 conditions. However, at day 28 and day 45, more AGEs were produced under the wa 0.75 condition. As indicated in Figure 2a and b, the inhibition of rosemary extract against AGEs formation was in a dose-dependent manner. Compared with the control group, RE3 reduced approximately 52.10% and 32.38% of AGEs at day 28 and day 56, respectively, under the wa 0.75 condition (Figure 2a). Similarly, RE3 reduced about 39.50% and 34.94% of AGEs at day 28 and day 56, respectively, under the wa 0.56 condition (Figure 2b). Generally, the Maillard reaction is relatively more active in the the wa range of 0.4–0.8, due to a dual effect of water. Under wa 0.75, the mobility of reactants was high, because the reactants became concentrated and this limited their diffusion under wa 0.56 [25]. Thus, higher level of AGEs was determined under wa 0.75 than wa 0.56.

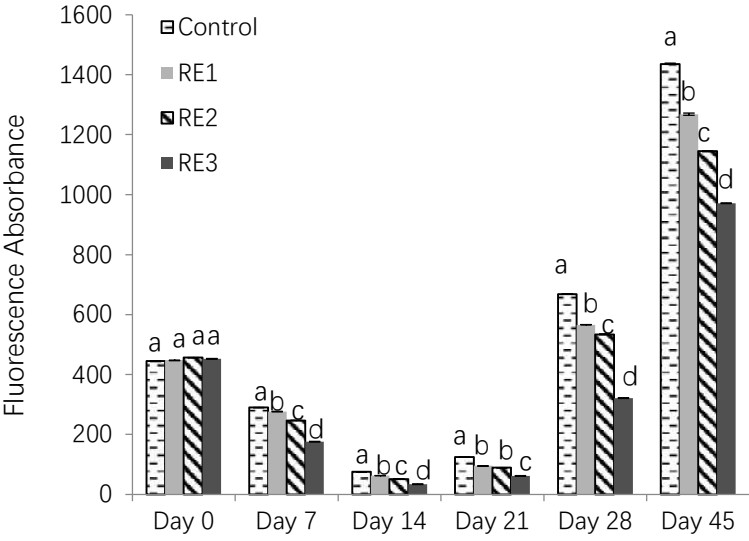

(a)

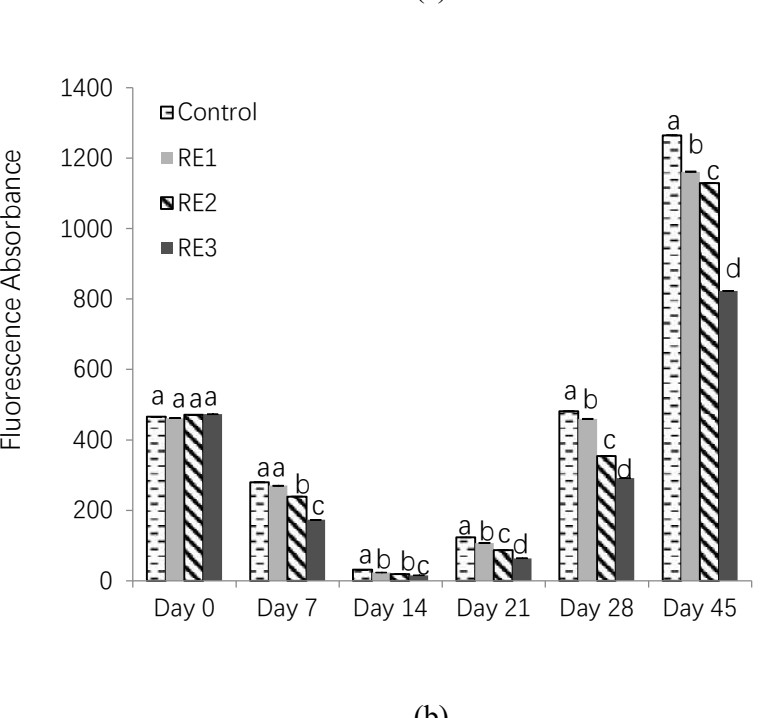

(b)

**Figure 2.** Fluorescence absorbances of AGEs in control and rosemary extract treatments under (a) wa 0.75 and (b) wa 0.56 conditions (RE1, 1%; RE2, 2%; RE3, 4%).

The CML accumulated with the increasing storage time under both of the conditions (Figure 3a and b). The inhibition of CML and AGEs may be due to the phenolic compounds in the rosemary extract. For example, rosemarinic acid was capable of inhibiting the formation of both total fluorescent AGEs and nonfluorescent carboxymethyllysine (CML) in a glucose-casein glycation model reported by Zhang [26]. There was 46.5% to 75.2% of CML reduction of with addition of carnosic acid [22]. Caffeic acid was found to reduce approximately 80% of CML at a fortification level

of 2.0 g in 100g bread dough [27]. In the study by Silván, the addition of ferulic acid reduced the formation of CML and fluorescent AGEs in vitro by nearly 90% by blocking free amino groups [28]. In addition, ellagic acid and protocatechuic acid demonstrated the antiglycative effects on the renal production of CML [29]. Those phenolics have been evidenced to have the potential to trap reactive dicarbonyl species through their benzene rings with one or more hydroxyl groups, thus, forming relatively nontoxic adducts, therefore inhibiting AGEs or CML [30]. Lastly, carnosic acid was proven to reduce AGEs and CML formation via pathways of conjugation with AGEs/CML precursors glyoxal (GO) and methylglyoxal (MGO) [31].

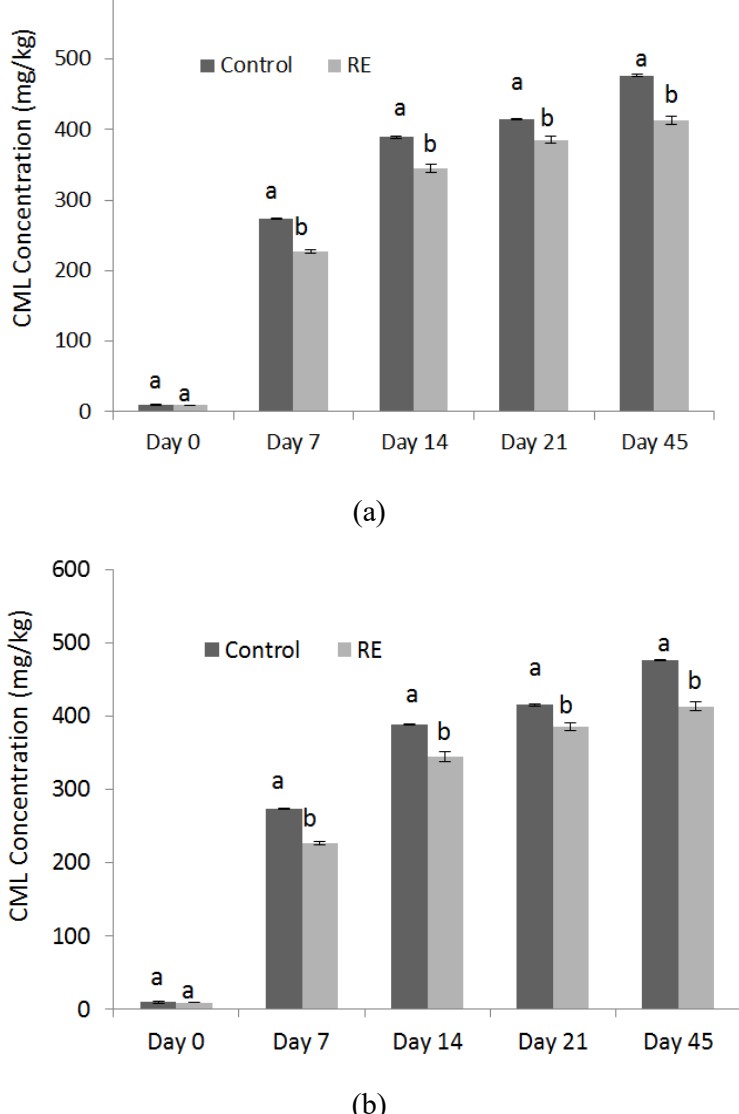

**Figure 3.** Concentrations of CML formed in control and RE treatment during storage under wa (a) 0.75 and (b) 0.56 conditions.

Furthermore, the profiles of glycated $\alpha$-lactalbumin (LA), $\alpha$-lactoglobulin (LG-A), and $\beta$-lactoglobulin (LG-B) in WPI protein were analyzed by LC-MS to investigate the modification of their structure during storage day 1, 3, 5, 7, and 21 under wa 0.75 condition. The molecular weights of LA, LG-A, and LG-B were 14177, 18362, and 18,277 Da, respectively (Figure 4a). The longer incubation time induced a higher degree of protein glycation which is indicated by the formation of high molecular weight derivatives of LG-A, LG-B, or LA. As shown in Figure 4b and c, there is no

significant difference between the control and RE3 treatment after day 1, since both of them produced high molecular weight derivatives of LG-A, LG-B, and LA with 4 to 8 glucose (G) molecules bound. After day 3, the order of the three highest abundance of glycated LG-A in the control was LG-A + 10G (20144 Da) > LG-A + 9G (19982 Da) > LG-A + 11G (20306 Da), while it was LG-A + 9G (19982 Da) > LG-A + 10G (20144 Da) > LG-A+8G (19822 Da) in the RE3 treatment (Figure 4d and e). At day 5, higher molecular weight glycated LG-A, LG-A + 13G (20630 Da) and LG-A + 14G (20794 Da) were observed in the control rather than RE3 treatment (Figure 4f and g). At day 7, the highest abundance of glycated protein was LG-B + 12G and LG-A + 12G in both the control and RE3 treatment, respectively (Figure 4h and i). However, after 21 days, multiple glycated derivatives occurred in both the control and RE3 treatment (Figure 4j & 4k). This indicates that the glycation of WPI protein was reduced by the rosemary extract within 21 days under the wa 0.75 condition. In the study of Miroliaei, Melissa officinalis L. extract which also contained in high levels of rosemarinic acid, which demonstrates the protection of BSA protein glycation. This was due to rosemarinic acid's potential to conceal the glycation sites and prevent cross β-structure formation in protein [32]. Besides the activity from multiple phenolic compounds, herb extracts are also confirmed to retard AGE-induced toxicity by the suppression of receptor signaling pathways [32].

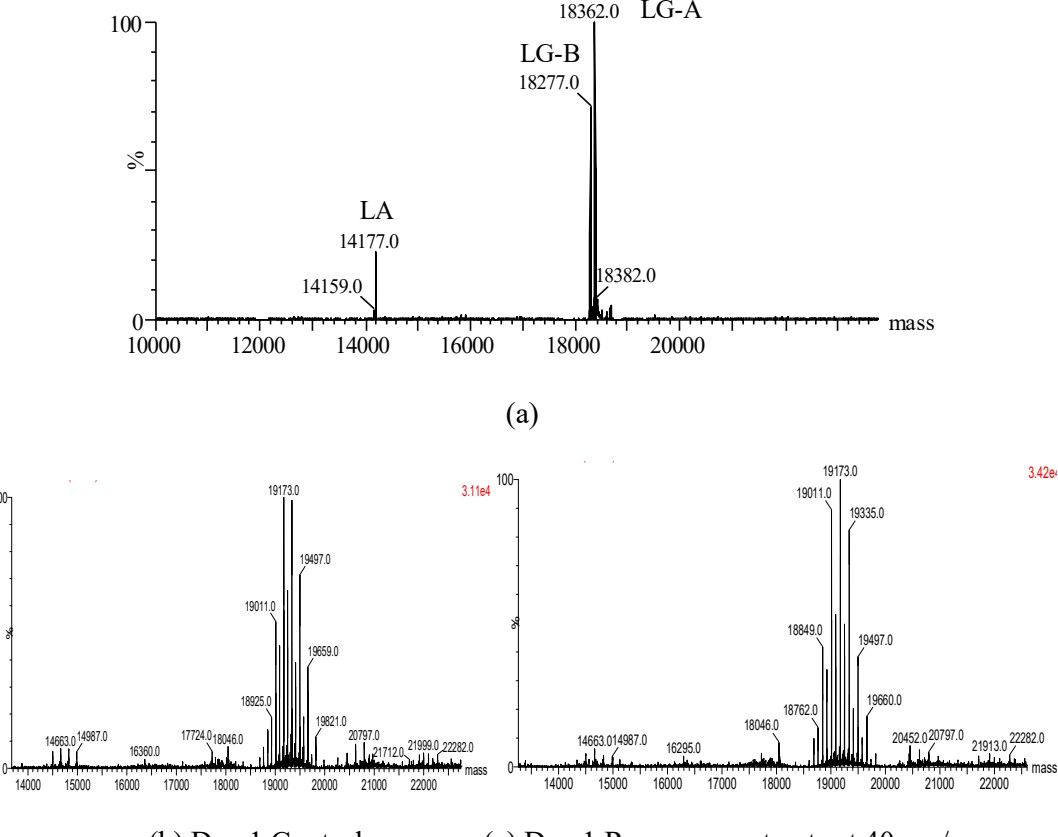

(a)

(b) Day 1 Control          (c) Day 1 Rosemary extracts at 40 mg/g

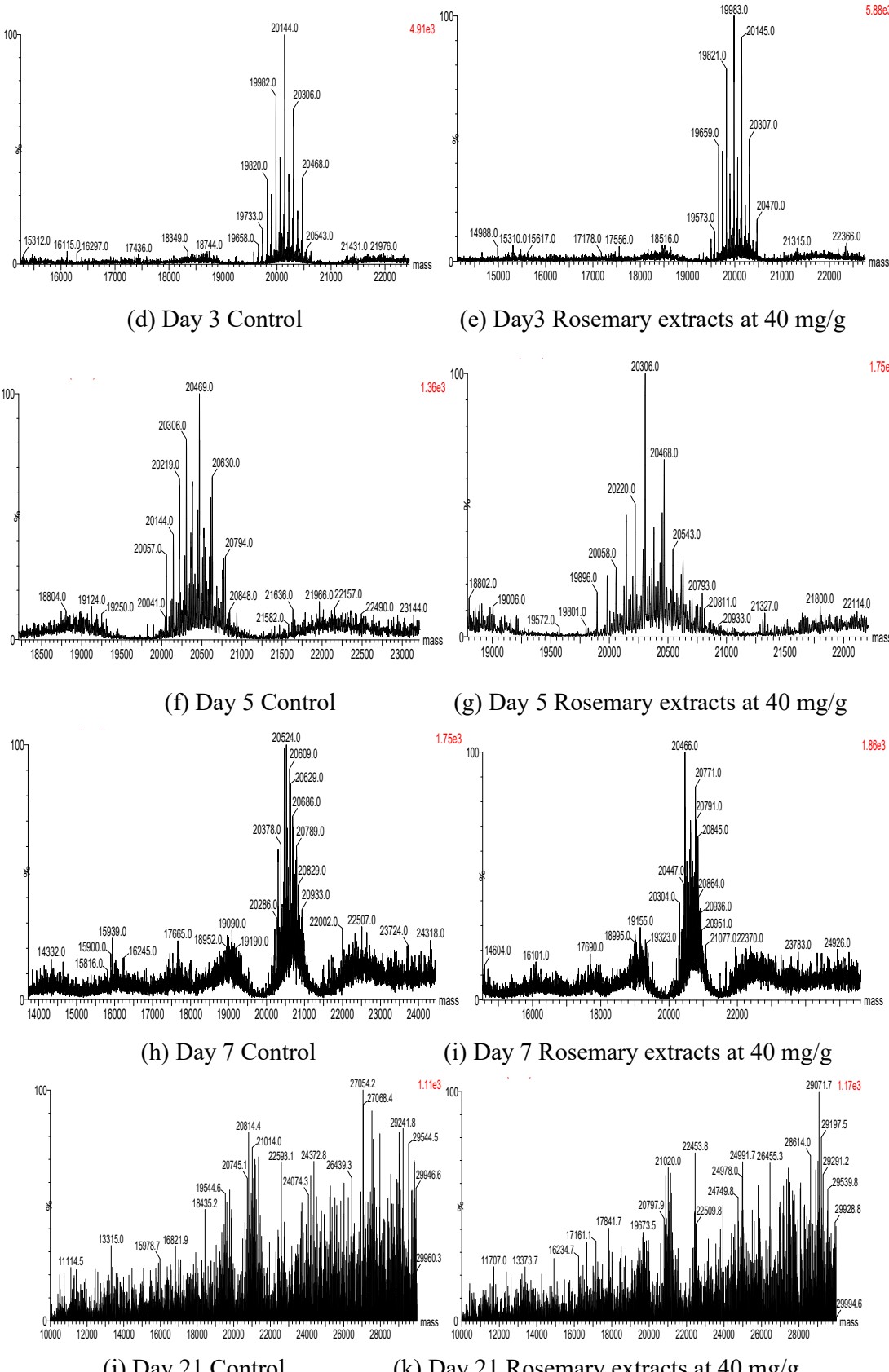

**Figure 4.** Mass spectrums of protein glycation in control and RE treatment during storage.

### 3.3. Effect of rosemary extract on proliferation of breast cancer cell line MCF-7

In this study, the effect of rosemary extract on the proliferation of cancer cells was investigated by treating MCF-7 cell lines with doses at 0.025, 0.050, 1.000, and 2.000 mg/mL for 96 h. As shown in Figure 5, the survival rate of MCF-7 cells decreased with the increase of concentrations of rosemary extract. Approximately 90.12% and 68.53% of MCF-7 cells survived at RE concentrations of 0.125 and 0.25 mg/mL, respectively (Figure 5). However, it significantly dropped to 13.57% at the RE concentration of 0.5 mg/mL (Figure 5) and reached 6.02% for RE of 1.0 mg/mL and 2.16% for RE of 2.0 mg/mL (Figure 5). The IC50 value of RE was 0.317 mg/mL. The inhibition of proliferation in MCF-7 cells could result from the induction of apoptosis by RE. Thus, TUNEL and DAPI co-staining assay was used to determine DNA fragmentation and nuclear condensation as apoptotic markers. Apoptosis is defined by a series of characteristic changes in cell morphology during cell death [33]. Morphological changes in MCF-7 cells treated with RE were observed. Control cells maintained a typical intact appearance, nevertheless, RE treated cells showed morphological alterations such as cellular shrinkage, cell monolayer area reduction, less adherence, and floating shapes (Figure 6).

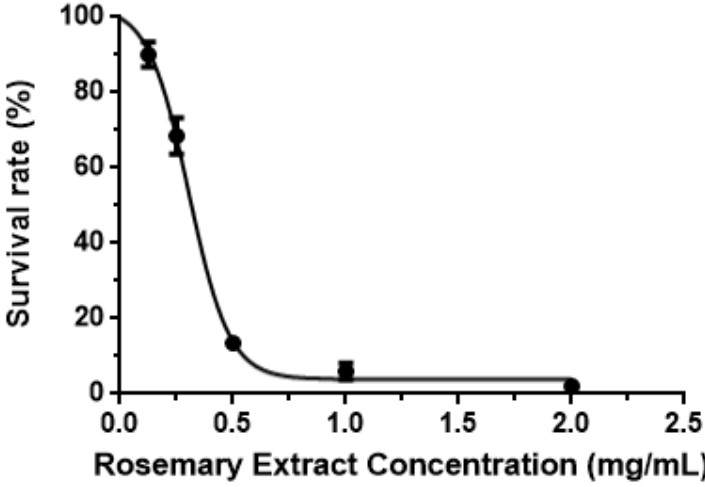

**Figure 5.** Survival rates of MCF-7 cancer cells treated with different levels of RE.

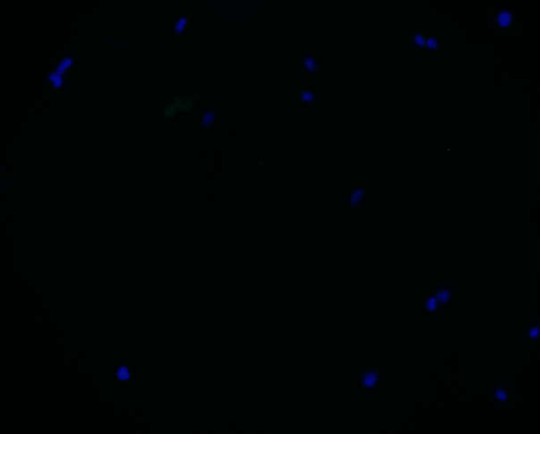

(a) Control.

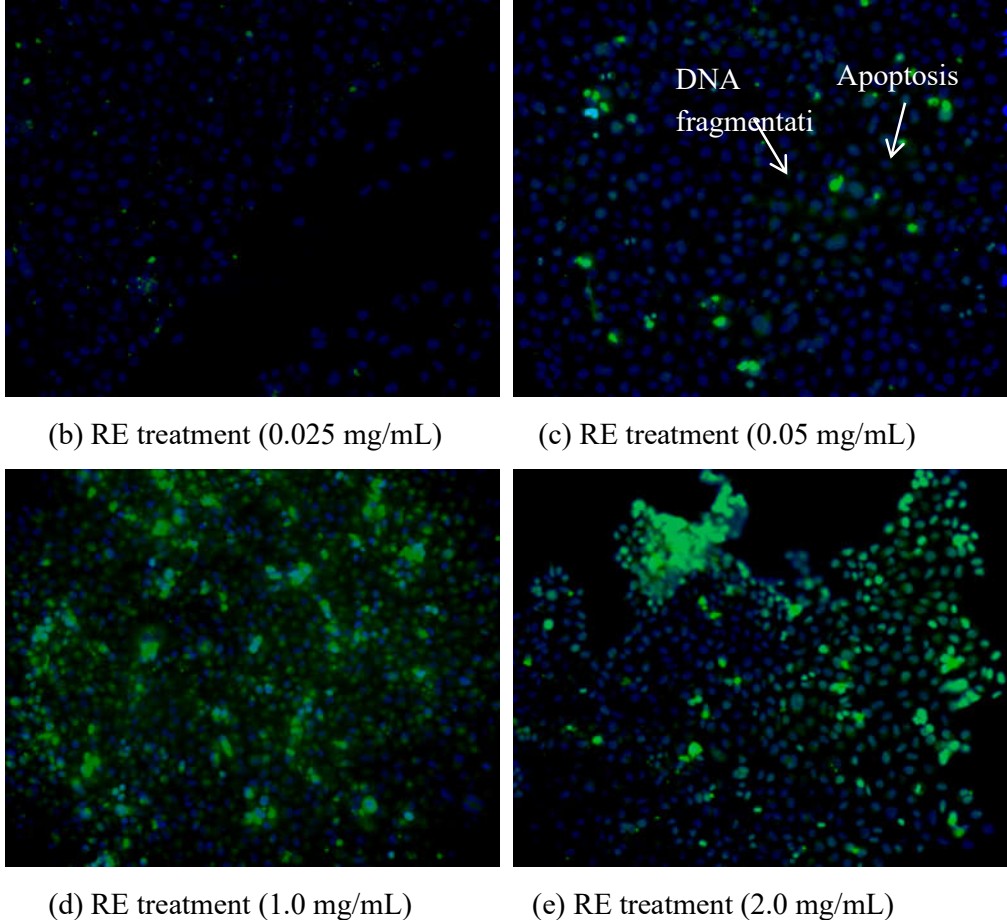

(b) RE treatment (0.025 mg/mL)  (c) RE treatment (0.05 mg/mL)

(d) RE treatment (1.0 mg/mL)  (e) RE treatment (2.0 mg/mL)

**Figure 6.** Images of DAPI staining (green color) and TUNEL (blue color) to indicate the MCF-7 cell apoptosis.

The efficient inhibitory performance of RE might be due to the abundance of phenolics. For example, a positive correlation between the number of unsubstituted hydroxyl groups (free OH), and the anti-proliferation capability was found in the study of Berdowska [34]. It has been reported that hydroxyl groups, especially in the form of ortho-diphenol moieties in rosemarinic acid and carbonyl functions (COOH) in carnosic acid could enhance the cytotoxic effects on the MCF-7 cell [34]. The related mechanism might lead one to speculate that the increased toxicity was contributed by ortho-diphenols through the chelation and reduction of transition metals (Fe, Cu, and Zn) which accelerate the cytotoxic effects on the MCF-7 cell [35]. A similar result has been reported in the study by Jaglanian, in which the anti-proliferative activity of rosemarinic acid was also observed in breast cancer cells by the induction of mitochondrial-mediated cancer cell apoptosis [36]. Another hypothetical mechanism is the interactions between phenolics and cellular proteins. The phenol and carbonyl groups of phenolics and amine or amide residues of a protein easily generate soluble or insoluble complexes via the formation of hydrogen bonds [35]. However, those interactions and the related products might affect and destroy important cellular structures. Additionally, carnosic acid could also contribute to the anti-proliferation capability of RE, since it accounts for approximately 24% of the total phenolics. The mechanism of carnosic acid was based on the induction of the G2/M phase of the cell cycle which led to cancer cell apoptosis [37]. These observations suggest that RE was effective in inducing apoptosis in MCF-7 cells.

### 3.4. Effect of rosemary extract on oxidative stress in diabetic rats

Reactive oxygen species (ROS) are produced during normal cellular metabolism [38]. They are highly reactive molecules and excessive amount of ROS can cause oxidative stress and damage cell structures [38]. Generally, the formation of ROS is regulated by an antioxidant system with an enzymatic-related antioxidant, such as superoxide dismutase (SOD), being involved [39]. Compared with the normal control (NC) (0.42 ± 0.02 U/mL), the increased activity of SOD (0.54 ± 0.04 U/mL) in the diabetes control (DC) could be induced by the accumulation of superoxide anions (Table 2). It is a defense mechanism that protects pancreatic β-cells from the adverse effect of increased superoxide production [40]. However, the SOD activity in the rosemary extract-treated diabetes group (DRE) (0.64 ± 0.03 U/mL) was significantly higher than DC and NC (Table 2). This correlates with a study that pomegranate extracts had the ability to ameliorate oxidative stress in plasma STZ-NA-induced diabetic rats, as evidenced by the improved antioxidant enzymatic status [39].

**Table 2.** Reactive oxygen species (ROS) in the experimental rats.

|  |  | Normal Control | Diabetes Control | DRE Treatment |
|---|---|---|---|---|
|  | T-SOD (U/mL) | 0.42 ± 0.02a | 0.54 ± 0.04b | 0.64 ± 0.03c |
| ROS | T-AOC (U/mL) | 2.19 ± 1.44a | 1.65 ± 0.10a | 6.13 ± 0.82b |
|  | MDA (nmol/mL) | 6.74 ± 0.49a | 8.43 ± 0.53c | 7.15 ± 0.26b |

T-SOD: total superoxide dismutase; T-AOC: total anti-oxidation capacity; MDA: malondialdehyde; DRE: diabetic group with rosemary extract treatment

Lipid peroxidation in diabetic rats could generate toxic products, inactivating membrane bound enzymes and damaging cell proteins through the direct free radical attack or chemical modification by its end products, such as malondialdehyde (MDA) [41]. Thus, as an important marker of lipid peroxidation, serum MDA significantly increased from 6.74 ± 0.49 nmol/L in NC to 8.43 ± 0.53 nmol/L in DC, due to the breakdown of hydrogen peroxide to generate hydroxyl radical. Nevertheless, it was reduced to 7.15 ± 0.26 nmol/L in DRE (Table 2). The inhibitory ability of RE could contribute to the $Fe^{2+}$ chelating and radicals scavenging capability of the extract, thus preventing the initiation of oxidative damage [42]. This is in accordance with Ahmadvand's study, which showed that a decrease was observed in MDA formation at the rates of 31.8%, 36.7%, and 50.3% by pure rosemarinic acid in a dose-dependent manner [43]. Additionally, Sheweita has reported that ethanoic extracts of plant Alhagi maurorum ameliorated the oxidative stress by decreasing the MDA level [44].

Due to lipid auto-oxidation, free radicals were formed disproportionately and antioxidant defenses were consumed, leading to cellular function disruption and oxidative damage to membranes [45]. The T-AOC could reflect overall cellular endogenous antioxidative capability [46]. The activities of T-AOC in DC (1.65 ± 0.10 U/mL) was significantly lower in NC (2.19 ± 1.44 U/mL), due to oxidative stress and inflammatory reaction (Table 2), while RE improved the serum T-AOC to 6.13 ± 0.82 U/mL. As reported in the studies of Shen, phenolic compounds such as rosemarinic acid and caffeic acid had the capability of scavenging free radicals [16]. Thus, the phenolics in RE could inhibit ROS and ameliorate the oxidative stress which helped to mediate the T-AOC activity in diabetic rats. Similar results have shown that various herbs such as Ilex paraguariensis and panax notoginseng were able to improve plasma T-AOC activity [47,48].

### 3.5. Effect of rosemary extract on blood glucose in diabetic rats

Generally, most dietary carbohydrates are broken down to monosaccharides in the upper gastrointestinal tract and then are absorbed in the circulation [49]. The increased glucose concentration in blood stimulates insulin secretion from β-cells in the pancreas, in response, insulin mediates the uptake of glucose [49]. If the pancreas function is suppressed, a resistance of insulin secretion will lead to various metabolic syndromes, such as hyperglycemia [49]. Thus, in this study, the blood glucose levels of the NC, DC, and DRE groups were monitored for 6 weeks. Table 3 indicates that NC had a stable blood glucose with a range of 5.07 ± 0.38 to 6.17 ± 0.56 mmol/L throughout 6 weeks of the experiment. However, DC and DRE had higher blood glucose levels than NC. The reason is because Streptozotocin-induced rats suffer from the selective destruction of

pancreatic insulin secreting β-cells, which make cells less active, eventually leading to poor glucose utilization by tissues [50]. From week 1 to 3, there was no significant anti-hyperglycemia effect of RE on the diabetic rat (Table 3). However, at week 4, approximately 30.93% of the blood glucose level was inhibited in treatment DRE (14.69 ± 2.11 mmol/L) as compared with DC (21.27 ± 1.83 mmol/L) (Table 3). The inhibitory percentage was approximately 25.56% and 21.00% at week 5 and week 6, respectively (Table 3). The possible mechanism was that pancreatic β–cells and insulin receptors were activated by the phenolics in RE, which, in turn, modulated the blood glucose level [51]. This also suggests that the phenolics could improve glucose uptake in peripheral tissues including muscle, adipocytes, and kidney, by activating the signaling pathways [51]. Additionally, those phenolic compounds might inhibit hepatic glycogen degradation and lead the liver to revert to its normal homeostasis through gluconeogenesis and glycogenesis [52]. Therefore, the results demonstrated that RE could decrease the elevated blood glucose level in the diabetic rat.

**Table 3.** Blood glucose levels in the experimental rats.

|  |  | Normal Control | Diabetes Control | DRE Treatment |
|---|---|---|---|---|
| Blood Glucose | 1W (mmol/L) | 5.43 ± 0.50a | 10.42 ± 1.33c | 8.54 ± 0.58b |
|  | 2W (mmol/L) | 5.07 ± 0.38a | 10.64 ± 0.96b | 11.31 ± 2.19b |
|  | 3W (mmol/L) | 5.16 ± 0.34a | 14.37 ± 1.21b | 13.19 ± 1.57b |
|  | 4W (mmol/L) | 5.18 ± 0.47a | 21.27 ± 1.83c | 14.69 ± 2.11b |
|  | 5W (mmol/L) | 5.35 ± 0.35a | 24.88 ± 2.16c | 18.52 ± 1.70b |
|  | 6W (mmol/L) | 6.17 ± 0.56a | 27.81 ± 3.54c | 21.97 ± 2.05b |

W: Week; DRE: diabetic group with rosemary extract treatment

*3.6. Effect of rosemary extract on lipid profile and liver function of diabetic rats*

Cholesterol (CHO) is an essential substance involved in the maintenance of cell membranes, the biosynthesis of vitamin D, and hormones. However, if CHO is above a normal level, it will result in atherosclerosis [53]. Similarly, triglycerides (TG) help store excessive energy in the biological system. Nevertheless, a high level of TG increases the risks of developing coronary artery diseases, such as heart attack or stroke, especially for diabetic patients [54]. High density lipoproteins (HDL) promote vascular health by the removal of cholesterol from the peripheral tissues and reduce macrophage accumulation [55]. On the contrary, low-density lipoproteins (LDLs) are the classic antagonists of the circulatory system, associated with atherosclerosis [55]. Thus, diabetic dyslipidemia is usually characterized by elevated cholesterol, triglycerides and LDL-C, as well as decreased high-density lipoprotein cholesterol (HDL-C) in comparison with healthy individuals. In this study, the effect of RE on the lipid profile, including CHO, TG, HDL-C, LDL-C, in diabetic rats was investigated. As shown in Table 4, DC has a significantly higher level of LDL (0.71 ± 0.14 mmol/L) than NC (0.37 ± 0.01 mmol/L), However, the treatment DRE reduced the LDL value to 0.49 ± 0.14 mmol/L. This ameliorative effect of RE could be due to the lipid-lowering potential of abundant phenolics in RE, which suppressed the activity of the hepatic LDL receptor site [56]. A slight reduction of plasma TG in DRE was observed, compared with DC, which was associated with a decrease of the LDL fraction. A similar trend was discovered in the CHO level. Results showed that it was relatively lower in DRE treatment than in DC (Table 4). This was due to the CHO synthesis and metabolism being suppressed by RE via the suppression of hydroxymethyl glutaryl-CoA (HMG-CoA) reductase and acyl CoA cholesterol/acyl transferase (ACAT), respectively [57,58]. Furthermore, phenolic acids such as carnosol, carnosic acid, and rosemarinic acid have been reported to inhibit endothelial cell-mediated oxidation of LDL in human aortic endothelial cells (HAEC), downregulated serum TG, and CHO [59,60]. Thus, RE can be absorbed, metabolized, and biologically active in the biological system and has the potential of maintaining the lipid profiles in diabetic rats.

**Table 4.** Lipids profiles and liver function parameters in the experimental rats.

|  |  | Normal Control | Diabetes Control | DRE Treatment |
|---|---|---|---|---|
| **Blood lipid** | **HDL-C (mmol/L)** | 1.45 ± 0.32a | 0.92 ± 0.22a | 1.37 ± 0.02a |
|  | **LDL-C (mmol/L)** | 0.37 ± 0.01a | 0.71 ± 0.14ab | 0.49 ± 0.14a |
|  | **TG (mmol/L)** | 0.80 ± 0.05a | 1.35 ± 0.08c | 1.21 ± 0.05b |
|  | **T-CHO (mmol/L)** | 2.48 ± 0.42a | 2.97 ± 0.12a | 2.63 ± 0.17a |
| **Liver Function** | **ALP (U/100mL)** | 43.81 ± 3.64a | 366.95 ± 10.99b | 349.25 ± 7.21b |
|  | **GOT (U/L)** | 47.34 ± 2.42a | 64.93 ± 2.35c | 50.53 ± 2.28b |
|  | **GPT (U/L)** | 37.37 ± 1.23a | 64.07 ± 3.29c | 50.61 ± 1.67b |

HDL-C: high-density lipoprotein-cholesterol; LDL-C: low-density lipoprotein-cholesterol; TG: triglycerides; T-CHO: total cholesterol; ALP: alkaline phosphatase; GPT: glutamate pyruvate transaminase; GOT: glutamate oxaloacetate transaminase; DRE: diabetic group with rosemary extract treatment

The liver is a primary organ involved in blood glucose regulation by glycogenesis, glycogenolysis, and gluconeogenesis [61]. Phosphate monoesterase and alkaline phosphatase (ALP) are widely found in body fluids [61]. ALP in serum mainly comes from the liver. Serum glutamate pyruvate transaminase (GPT) and glutamate oxaloacetate transaminase (GOT) are essential aminotransferases widely distributed in the cell membrane, cytoplasm, and mitochondria [62]. Table 4 shows that there was a significantly increase of ALP activity in DC (366.95 ± 10.99 U/100mL) compared with NC (43.81 ± 3.64 U/100mL). Treated DRE could slightly control ALP activity at a level of 349.25 ± 7.21 U/100mL which indicated that it was difficult for RE to generate a reversible effect on ALP in diabetic rats (Table 4). The elevation of GOT and GPT activities of plasma was mainly attributed to the leakage of these enzymes from the liver cytosol into the blood stream. Thus, they could be used as an evaluation indicator on the damage of the liver function [63]. In this study, the GOT level in NC was 47.34 ± 2.42 U/L, which was approximately 27.10% less than DC (64.93 ± 2.35 U/L) (Table 4). After the oral administration of RE, the hepatoprotective effect was observed in the treated DRE whose GOT activity normalized to 50.53 ± 2.28 U/L (Table 4). Similarly, RE helped reduce GPT activity to 50.61 ± 1.67 U/L compared with DC (64.07 ± 3.29 U/L) (Table 4). The results demonstrate that phenolics in RE could exhibit hepatocytes protective performance based on the mechanism of maintaining the functional integrity of the hepatocyte membrane [64]. The antioxidant mechanism on inhibition of oxidative stress was also involved in protecting against acute hepatic lesions, as indicated in the study by Sugiyama [65]. Since RE contain various phenolic compounds with relatively high antioxidant activities, they could protect the liver cells by scavenging free radicals and suppressing oxidative stress. Similar results have been reported in the studies of Jung et al. and Nwanna et al., which state that the Rhus verciflua and T. occidentalis leaf extract could improve GOT, GPT, and ALP activity in diabetic or liver-damaged rats [57,66].

## 4. Conclusions

In this study, RE demonstrated an inhibitory effect on AGEs, CML, and protein glycation formation. In addition, RE had the potential of inhibiting breast cancer cell proliferation. The risk of the diabetic rat was also proven to be lowered by the RE treatment via mediating blood glucose, serum malondialdehyde (MDA), cholesterol (CHO), triglycerides (TG), low-density lipoproteins (LDLs), anti-oxidation capacity (T-AOC), and superoxide dismutase (SOD) activity. Additionally, the liver function of the diabetic rat was improved by RE, through maintaining alkaline phosphatase (ALP), glutamate pyruvate transaminase (GPT), and glutamate oxaloacetate transaminase (GOT) activities.

**Author Contributions:** Conceptualization, D. Xiao; formal analysis, Y.Yang; methodology, L. Zheng; software, B. Ai; resources, J. Han; data curation, X. Zheng; writing—review and editing, Y. Shen; supervision, Z. Sheng. All authors have read and agreed to the published version of the manuscript.

**Funding:** This research was supported by the National Natural Science Foundation of China (31772096), the Key Technologies Research and Development Program of Hainan (ZDXM2014104 and SF201441), Central

Public-interest Scientific Institution Basal Research Fund for Innovative Research Team Program of CATAS (NO. 17CXTD-05, NO.1630092019001).

**Acknowledgements:** We thanks to the National Natural Science Foundation of China (31772096), the Key Technologies Research and Development Program of Hainan (ZDXM2014104 and SF201441), Central Public-interest Scientific Institution Basal Research Fund for Innovative Research Team Program of CATAS (NO. 17CXTD-05, NO.1630092019001).

**Conflicts of interest:** The authors declare no conflict of interest.

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
