# Peer review of "Rosemary Leaf Extract Inhibits Glycation, Breast Cancer Proliferation, and Diabetes Risks"

_applsci, doi:10.3390/app10072249_

Round 1
Reviewer 1 Report
In the present study, the extract form Rosemary leaves was studied in terms of its inhibitory effect on AGEs and CML formation, and protein glycation. Finally, its inhibitory effects in these areas were clearly presented.
What is more, its effect on breast cancer cell proliferation was estimated in MCF-7 cells, and the potential of inhibiting breast cancer cell proliferation was shown.
In addition, the Authors studied the effect of the extract on blood glucose, cholesterol, lipoproteins, and the antioxidative defence system, as well as on the markers of the liver function in streptozotocin-induced diabetic rats.
To me, the most interesting part concerns the abilities of Rosemary extract in inhibiting protein glycation, AGEs and CML formation in sugar-protein rich intermediate-moisture food. Also, its inhibitory effect on non-invasive hormone-dependent breast cancer cell could be of the great interest. In this area, scientific novelty as well as the significance of the results seem to be proved. Other results are interesting, however, they are similar to many others present in the literature concerning phenolic compounds from Lamiaceae.
In summary, the manuscript could be interesting for a reasonable number of scientists working on new treatments of metabolic and cancer diseases.
However, the manuscript could be improved in some parts. To me, more explanation about two different water activity conditions in sugar-protein rich intermediate-moisture food could be included. Also, some discussion about the differences observed in these conditions could be added.
Author Response
The authors all thank the reviewer for pointing out the significance of the study.
The answers to the reviewer’s suggestions were listed below:
- Explanation about two different water activity conditions has been added.
- The discussion about the differences observed under different water activities were added.
Reviewer 2 Report
- This manuscript aimed to look at the RE abilities to inhibit nvasive hormone-dependent breast cancer cell (MCF-7). This topic has been discussed in a previous paper [González‐Vallinas, M., Molina, S., Vicente, G… (2014). Modulation of estrogen and epidermal growth factor receptors by rosemary extract in breast cancer cells. Electrophoresis, 35(11), 1719-1727], the authors are suggested to strengthen the novelty of this study. The English language is well written.
- If the focus of this manuscript is to use the rosemary extract (RE), we, unfortunately, know little about it, such as its composition in general… The authors should conduct a composition analysis, or at least, seek the data from the literature.
- The centrifuged at 4000 g, it should be 4000 xg and this apply to the rest of the manuscript.
- Please keep the format of the tables and figures consistent.
Author Response
The authors all thank the reviewer for pointing out the rationality of the study.
The answers to the reviewer’s suggestions were listed below:
- This manuscript not only investigated the anticancer potential of rosemary extract but also demonstrated its efficiency on antiglycation. Thus, this manuscript is a more comprehensive study for evaluation the bioactive functions of rosemary extracts. The novelty of the study has been strengthened in the introduction section.
- The phenolic profile was shown in Table 1 and the typical chromatogram has been added as Figure 1.
- The centrifuged speed has been changed to 4000 xg and applied throughout the manuscript.
- The format of tables and figures has been modified to keep consistency.